Methods

# Refining the genetic risk of breast cancer with rare haplotypes and pattern mining

William Letsou[1,2], Fan Wang[2], Wonjong Moon[2], Cindy Im[3,4], Yadav Sapkota[2], Leslie L Robison[2], Yutaka Yasui[1,4]

**Hundreds of common variants have been found to confer small but significant differences in breast cancer risk, supporting the widely accepted polygenic model of inherited predisposition. Using a novel closed-pattern mining algorithm, we provide evidence that rare haplotypes may refine the association of breast cancer risk with common germline alleles. Our method, called Chromosome Overlap, consists in iteratively pairing chromosomes from affected individuals and looking for noncontiguous patterns of shared alleles. We applied Chromosome Overlap to haplotypes of genotyped SNPs from female breast cancer cases from the UK Biobank at four loci containing common breast cancer-risk SNPs. We found two rare (frequency <0.1%) haplotypes bearing a GWAS hit at 11q13 (hazard ratio = 4.21 and 16.7) which replicated in an independent, European ancestry population at $P < 0.05$, and another at 22q12 (frequency <0.2%, hazard ratio = 2.58) which expanded the risk pool to noncarriers of a GWAS hit. These results suggest that rare haplotypes (or mutations) may underlie the "synthetic association" of breast cancer risk with at least some common variants.**

## Introduction

Genome-wide association studies (GWAS) have identified hundreds to thousands of single-nucleotide polymorphisms (SNPs) robustly associated with breast cancer risk and other complex phenotypes (Michailidou et al, 2017; Zhang et al, 2020). Polygenic risk scores derived from common and genome-wide variants can differentiate women's breast cancer risk by up to several fold (Ge et al, 2019; Mavaddat et al, 2019; Mars et al, 2020). The widely accepted polygenic model of genetic risk (Pharoah et al, 2002) contrasts with the hypothesis that rare variants are responsible for the elevated risk with which GWAS SNPs are but "synthetically" associated (Dickson et al, 2010; Anderson et al, 2011; Wray et al, 2011). According to the latter model, many complex diseases are characterized by genetic heterogeneity (McClellan & King, 2010), or by a small pool of individuals having private, rare mutations each associated with a large increase in risk. We have recently argued on the basis of high-quality data from several Nordic cancer registries (Möller et al, 2016) that the elevated breast cancer risk to twins of breast cancer cases is largely because of rare variants or haplotypes of large effect (Yasui et al, 2023). It is imperative to reconcile the polygenic and synthetic models to better understand the etiology of this and other diseases.

Haplotype analysis has long been used as a method to improve power in genetic association studies (Falk & Rubinstein, 1987; Schaid, 2004), but the idea that haplotypes themselves may tag or could themselves be the causal variants has received less attention. We recently explored the possibility in a genome-wide analysis that rare haplotypes with copy numbers on the order of tens per 10,000 chromosomes confer additional high risk for breast cancer (Wang et al, 2023). That study identified five robustly replicated rare haplotypes defined by genotyped SNPs through a sliding-window analysis, but was necessarily constrained by the requirement that haplotypes be *contiguous* regions of chromosome of various sizes, each possibly bearing a rare causal variant. The hypothesis that rare risk haplotypes could be *noncontiguous*, perhaps representing the interaction of common variants at different locations in a gene regulatory region, has to our knowledge not been previously considered. As haplotypes are combinations of alleles on one and the same chromosome, they are difficult to study in a systematic manner: for one, without phased genome sequencing, large numbers of chromosomes need to be computationally phased. More importantly, combinatorics quickly engenders a multiple-testing problem, as there are $3^m - 1$ possible haplotypes (both contiguous and noncontiguous) in every $m$-SNP window. Although most of these haplotypes will never appear in the population, identifying the ones to test for disease association requires new computational tools not previously used on a large-scale in genetic association studies.

Closed-pattern mining is a well-known technique in computer science used to find items that frequently occur together in multivariate data, including, for example, grocery items purchased together in supermarket transactions (Pasquier et al, 1999). Applied to market data, a pattern of goods is said to be *closed* if no item can

---

[1]Department of Biological & Chemical Sciences, New York Institute of Technology, Old Westbury, NY, USA   [2]Department of Epidemiology and Cancer Control, St. Jude Children's Research Hospital, Memphis, TN, USA   [3]Department of Pediatrics, University of Minnesota, Minneapolis, MN, USA   [4]School of Public Health, University of Alberta, Edmonton, Canada

Correspondence: wletsou@nyit.edu; yutaka.yasui@stjude.org

be added to it without diminution of the number of transactions in which the pattern appears, and the *closure* of any pattern is the shortest closed-pattern which contains it as a subset (Uno et al, 2004). Applied to haplotype analysis, a pattern of SNP-alleles is said to be closed if it is the longest pattern shared by a group of chromosomes, capturing one and the same set of chromosomes as certain shorter patterns and therefore obviating the need for exhaustive enumeration. Versions of closed-pattern mining (Pan et al, 2003; Terada et al, 2016; Relator et al, 2018; Yoshizoe et al, 2018) and a related approach called frequent pattern mining (Fang et al, 2012; Okazaki et al, 2021; Pounraja and Girirajan, 2022) have been applied in various genetic association studies to the learning of association rules between combinations of genes/genotypes and oligogenic disease; but to our knowledge none of these methods has been applied to haplotype analysis. As it is haplotypes—not genotypes—that are passed from parent to offspring, it is pertinent to look for the rare, possibly noncontiguous patterns of alleles that could underlie the genetic heterogeneity of inherited breast cancer risk.

In this article, we find using a new closed-pattern mining algorithm called Chromosome Overlap that rare haplotypes can refine the risk associated with "GWAS hits," common variants previously shown to be associated with breast cancer risk. Our method consists in iteratively overlapping pairs of chromosomes from affected individuals and looking for shared, noncontiguous haplotypes. We then compared the counts of (the closures of) these patterns in cases and controls on the hypothesis that the sharing of patterns by cases should be associated with their being cases. We

apply Chromosome Overlap to computationally phased data from female breast cancer cases and controls in the UK Biobank (UKBB) (Bycroft et al, 2018) to look for rare haplotypes in the vicinity of three of the strongest breast cancer (EFO_0000305) hits by *P*-value in the NHGRI-EBI GWAS Catalog (Sollis et al, 2023), including: rs2981578 on chromosome 10q26 in an intron of *FGFR2* (Meyer et al, 2008); rs554219 on chromosome 11q13 upstream of *CCND1* (French et al, 2013); rs4784227 on 16q12 in an intron of *CASC16* (Long et al, 2010; Ulder et al, 2010); and also a locus at 22q12 containing a rare haplotype identified by our previous genome-wide haplotype analysis (Wang et al, 2023). Subsequently, we replicated the 11q13 and 22q12 results in cases and controls from the Discovery, Biology, and Risk of Inherited Variants in Breast Cancer (DRIVE) study, lending support to the hypothesis that rare haplotypes, or the rare variants they tag, underlie the synthetic association of breast cancer risk with at least some GWAS hits.

# Results

### Discovery and replication of two rare haplotypes at the 11q13 locus

We applied Chromosome Overlap around three of the most strongly associated breast cancer risk SNP-alleles. For each of the three GWAS hits in Phase 1, we focused on a ~200-kb region containing the

**Table 1. Numbers of closed-patterns discovered at four loci during Chromosome Overlap, phases 1 and 2.**

| | Locus (no. of SNPs) | | | |
|---|---|---|---|---|
| | **10q26 (55)** | **11q13 (57)** | **16q12 (56)** | **22q12 (49)** |
| Iteration | | | *Phase 1* | |
| 0 | 2,343 | 2,520 | 2,858 | 4,221 |
| 1 | 91 (2,511,040[a]) | 121 (2,962,894[a]) | 246 (3,458,020[a]) | 127 (7,759,059[a]) |
| 2 | 1,509 | 3,207 | 8,294 | 3,040 |
| 3 | 42,368 | 105,316 | 138,303 | 93,751 |
| 4 | 454,025 | 461,915 | 122,982 | 297,225 |
| 5 | 87,598 | 15,291 | 108 | 4,108 |
| | **10q26 (158)** | **11q13 (92)** | **16q12 (195)** | |
| Iteration | | | *Phase 2* | |
| 0 | 5,543 | 699 | 3,452 | |
| 1 | 30 (15,353,437[a]) | 28 (241,811[a]) | 25 (5,954,654[a]) | |
| 2 | 401 | 342 | 300 | |
| 3 | 19,667 | 10,989 | 14,049 | |
| 4 | 503,027 | 132,560 | 881,903 | |
| 5 | 208,401 | 24,079 | 1,620,974 | |
| 6 | | | 16 | |
| 7 | | | 78 | |
| 8 | | | 474 | |
| 9 | | | 102 | |

[a]Total number of patterns before filtering.

GWAS hit and extracted haplotypes containing 50–60 UKBB-genotyped SNPs (see the Materials and Methods section). The first region selected was chr11:69,419,318–69,616,860, a range containing 57 genotyped SNPs about, but not including, the GWAS hit rs554219.

Table 1 shows that there were 2,520 unique contiguous haplotypes at the chr11 locus among the 18,022 chromosomes of UKBB breast cancer cases and 2,962,894 meta-chromosomes resulting from the initial pairwise overlap. We filtered the meta-chromosomes down to 121 with a Fisher's exact test $P$-value threshold of $P = 1.0 \times 10^{-9}$ to prevent a combinatorial explosion at subsequent iterations (see the Materials and Methods section). The 121 filtered meta-chromosomes resulted in a total of 585,850 closed-patterns within five iterations, after which point, no more closed-patterns were found.

Among the 585,850 noncontiguous closed-patterns and 2,520 original contiguous patterns, we evaluated the top 20,628 (with Fisher's exact $P < 1.0 \times 10^{-15}$) in a Cox proportional hazards model for association with breast cancer incidence rates (see the Materials and Methods section), after verifying that none of the original contiguous patterns met the inclusion threshold (minimum $P$-value: $2.4 \times 10^{-5}$). We found that each of the evaluated haplotypes had a Cox likelihood ratio test (LRT) $P$-value less than $1.0 \times 10^{-5}$, but that only one pattern remained after stepwise forward selection. This 19-SNP haplotype was reduced by recursive partitioning to 17

SNP-alleles (Table S1) without alteration of its frequency or breast cancer incidence hazard ratio (HR). As shown in Table 2, this haplotype, designated h1, was highly statistically significant in UKBB (and also DRIVE), but its association strength was on par with that of the GWAS hit. Furthermore, h1 was in LD ($r^2 = 0.78$, $D' = 1.00$) with its corresponding GWAS hit, rs554219[G]. We thus concluded that the common haplotype h1 was tagging largely the same pool of at-risk subjects as was rs554219[G].

Because we were interested in rare haplotypes that underlie GWAS hits, we hypothesized that there were rare subtypes of h1 in the immediate chromosomal vicinity which could be found by a conditional analysis of h1-bearing chromosomes. To test this hypothesis in Phase 2, we extracted SNPs in the topologically-associated domains (TADs) containing h1, reasoning that chromosomal interactions might mediate the risk associated with the putative rare haplotypes. Using the 3D Genome Browser (Wang et al, 2018) to predict domain boundaries, we extracted haplotypes in the region chr11:69,083,946–69,414,699 (92 SNPs) coinciding with the 5′ boundary of the TAD and the 5′ boundary of h1 (Fig S1) from 2,100 h1-bearing chromosomes of UKBB breast cancer cases. This region contained 699 unique contiguous haplotypes and 241,811 unique noncontiguous pairwise overlaps (Table 1).

Because of the greater number of SNPs. filtering had to be more stringent in Phase 2, with the additional requirement that a filtered meta-chromosome be the shortest member of a family with the same

**Table 2. Common-haplotypes at 10q26, 11q13, and 16q12, discovery and replication.**

| | HR/OR[a] | Cases chromosomes (per 10,000) | Controls chromosomes (per 10,000) | $P$[b] |
|---|---|---|---|---|
| | | 10q26:121,481,608–121,680,765 | | |
| Variable | | *UKBB discovery* | | |
| h1 | 1.30 | 7,976 (4,426) | 130,297 (3,787) | $3.7 \times 10^{-66}$ ($3.2 \times 10^{-65}$) |
| rs2981578[C] | 1.25 | 9,332 (5,178) | 158,551 (4,608) | $1.7 \times 10^{-51}$ ($2.3 \times 10^{-50}$) |
| | | *DRIVE replication* | | |
| h1 | 1.26 | 26,338 (4,380) | 19,393 (3,835) | $2.2 \times 10^{-75}$ ($3.0 \times 10^{-75}$) |
| rs2981578[C] | 1.22 | 31,504 (5,239) | 24,045 (4,755) | $1.7 \times 10^{-58}$ ($6.0 \times 10^{-58}$) |
| | | 11q13:69,419,318–69,616,860 | | |
| | | *UKBB discovery* | | |
| h1 | 1.26 | 2,100 (1,165) | 32,586 (947.1) | $1.5 \times 10^{-21}$ ($4.0 \times 10^{-21}$) |
| rs554219[G] | 1.24 | 2,568 (1,425) | 40,679 (1,182) | $3.0 \times 10^{-22}$ ($1.3 \times 10^{-21}$) |
| | | *DRIVE replication* | | |
| h1 | 1.21 | 6,825 (1,135) | 4,894 (967.9) | $3.1 \times 10^{-21}$ ($1.8 \times 10^{-19}$) |
| rs554219[G] | 1.21 | 8,802 (1,464) | 6,333 (1,254) | $7.9 \times 10^{-27}$ ($1.7 \times 10^{-24}$) |
| | | 16q12:52,486,414–52,645,181 | | |
| | | *UKBB discovery* | | |
| h1 | 1.27 | 4,595 (2,550) | 72,883 (2,118) | $1.2 \times 10^{-41}$ ($1.9 \times 10^{-41}$) |
| rs4784227[T] | 1.25 | 5,061 (2,808) | 81,448 (2,367) | $1.2 \times 10^{-40}$ ($2.6 \times 10^{-40}$) |
| | | *DRIVE replication* | | |
| h1 | 1.22 | 14,616 (2,431) | 10,582 (2,093) | $8.5 \times 10^{-43}$ ($7.9 \times 10^{-41}$) |
| rs4784227[T] | 1.23 | 16,971 (2,822) | 12,305 (2,434) | $1.8 \times 10^{-50}$ ($1.6 \times 10^{-48}$) |

[a]HR (hazard ratio) for UKBB, OR (odds ratio) for DRIVE.
[b]LRT (likelihood ratio test) $P$-value adjusted for age and 10 principal components of ancestry (Fisher's exact test $P$-value).

case/control frequency in UKBB (see the Materials and Methods section). At the chr11 locus, we kept 28 filtered haplotypes with h1-conditional Fisher's exact test $P < 1.0 \times 10^{-4}$ (among the h1 carriers only) after the first overlap, resulting in a total of 167,998 closed-patterns from Iteration 1 onward (Table 1). As in Phase 1, all closed-patterns at the chr11 locus were discovered within five iterations.

Among the 167,998 noncontiguous closed-patterns and 699 original contiguous patterns, we evaluated the top 246 (with h1-conditional Fisher's exact test $P < 1.0 \times 10^{-4}$ among h1 carriers only) in a Cox proportional hazards model for association with breast cancer incidence rates, after verifying that none of the original 699 contiguous haplotypes (minimum h1-conditional $P$-value $3.1 \times 10^{-3}$) met the inclusion threshold. At the chr11 locus, we found 106 risk-increasing closed-patterns with Cox-LRT $P < 1.0 \times 10^{-5}$ in the UKBB discovery analysis, of which 14 had $P < 0.05$ in the DRIVE replication analysis (Table S2). In the permutation analysis, no risk-increasing patterns were found at the $P < 1.0 \times 10^{-5}$ level, suggesting that all 106 discovered risk haplotypes were genuine, although the replication rate was only 13% (see the Discussion section).

We next asked whether the 14 risk haplotypes were not variants of the same risk haplotype. Using stepwise forward selection, we found that only two of the 14 risk haplotypes at the chr11 locus were independently associated with breast cancer risk. These haplotypes, designated h2 and h3 in Tables 3 and S3, were both rare (fewer than 5 and 1 copies per 10,000 chromosomes, respectively, in controls) and, when adjoined to 17-SNP h1, were 80 and 86 SNPs long. h2 and h3 were highly risk-increasing (HRs of 4.21 and 16.7) in the discovery analysis and in DRIVE (odds ratios [ORs] of 2.10 and 11.7).

### Features of the rare haplotypes

To investigate how h2 and h3 from the chr11 locus could be involved in breast cancer risk, we used the WashU Epigenome Browser (Li et al, 2022) to plot the locations and SNP-alleles of the haplotypes together with ENCODE Hi-C and ChIP-seq data (ENCODE Project Consortium, 2012; Davis et al, 2018) (Fig 1).

h2 and h3 were highly similar in terms of both their included SNPs and allele phases. Both haplotypes used most, but not all, of the 149 UKBB-typed SNPs in the region. Alternate alleles (red) appeared at discrete locations, most notably at the *MYEOV* locus (~69.3 Mb) and at the 5′ end (position ~69.13 Mb) of the TAD. According to the Hi-C data,

these two loci participate in a chromosomal loop and appear to physically interact with another locus in the vicinity of *CCND1*, the gene encoding the cell cycle regulatory molecule cyclin D1. The similarities of and differences between h2 and h3 suggest (1) that the haplotypes are distinct signals which are also distinct from contiguous haplotypes in the region, and (2) that there may be additional undetected versions of these haplotypes containing a common "backbone" of SNP-alleles.

We also observed coincident binding throughout the TAD of transcription factors relevant to breast cancer, including CTCF, FOXA1, ESRRA (a relative of the estrogen receptor ER), and MYC. We focused on these particular factors because (1) the CTCF–FOXA1–ER triplet is known to mediate cells' response to estrogen (Hurtado et al, 2011; Ross-Innes et al, 2011), and (2) according to the HACER database (Human ACTive Enhancer to interpret Regulatory variants) (Wang et al, 2019), MYC binds to predicted enhancers of *CCND1*. We observed colocalization of CTCF, FOXA1, and MYC at the 5′ end of the haplotype and again at a downstream region (~69.45 Mb) devoid of SNP-alleles. ESRRA, FOXA1, and MYC bind at the *MYEOV* location (~69.3 Mb and ~69.27 Mb) displaced from a CTCF peak in a location adjacent to—but not on top of—several alternative alleles in h2 and h3. We verified that in both of these "empty" regions, the UKBB data are lacking any typed SNPs. Finally, the original GWAS hit rs554219 exhibits the *MYEOV*-binding pattern and also forms a loop with the locus at 69.45 Mb. These binding data support the hypothesis that the haplotypes identified by Chromosome Overlap are picking out biologically relevant features of the *CCND1* locus.

### Application of Chromosome Overlap to two other GWAS hits

Having discovered two rare haplotypes that appeared to be genuine predictors of breast cancer risk, we asked if we could not find similar haplotypes at two other GWAS hits. The selected regions for these Phase 1 analyses were chr10:121,481,608–121,680,765 (55 SNPs about rs2981578, Fig S2) and chr16:52,486,414–52,645,181 (56 SNPs about rs4784227, Fig S3), the latter of which contained the GWAS hit as a typed SNP in the phased data. As shown in Table 1, these regions contained 2,343 and 2,858 unique contiguous chromosomes from UKBB breast cancer cases and 2,511,040 and 3,458,020 unique pairwise overlaps.

After adjusting the Fisher's exact test $P$-value thresholds, we found 585,591 closed-patterns at the chr10 locus starting from 91 meta-chromosomes with $P < 1.0 \times 10^{-20}$ and 269,993 at the chr16

**Table 3. Rare-haplotypes at 11q13, discovery and replication.**

| | HR/OR[a] | Cases chromosomes (per 10,000) | Controls chromosomes (per 10,000) | $P$[b] |
|---|---|---|---|---|
| | \multicolumn{4}{c}{11q13:69,083,946–69,414,699} | | | |
| Variable | \multicolumn{4}{c}{UKBB discovery[c]} | | | |
| h2 | 4.21 | 26 (14.4) | 116 (3.37) | $2.4 \times 10^{-7}$ ($8.9 \times 10^{-9}$) |
| h3 | 16.7 | 5 (3.3) | 6 (0.15) | $7.3 \times 10^{-6}$ ($5.6 \times 10^{-6}$) |
| | \multicolumn{4}{c}{DRIVE replication[c]} | | | |
| h2 | 2.10 | 52 (8.65) | 22 (4.35) | 0.025 ($6.9 \times 10^{-3}$) |
| h3 | 11.7 | 13 (2.16) | 1 (0.20) | $2.8 \times 10^{-3}$ ($5.0 \times 10^{-3}$) |

[a]HR (hazard ratio) for UKBB, OR (odds ratio) for DRIVE.
[b]LRT (likelihood ratio test) $P$-value adjusted for h1, age, and 10 principal components of ancestry (Fisher's exact test $P$-value).
[c]Haplotypes fitted jointly in UKBB and singly in DRIVE.

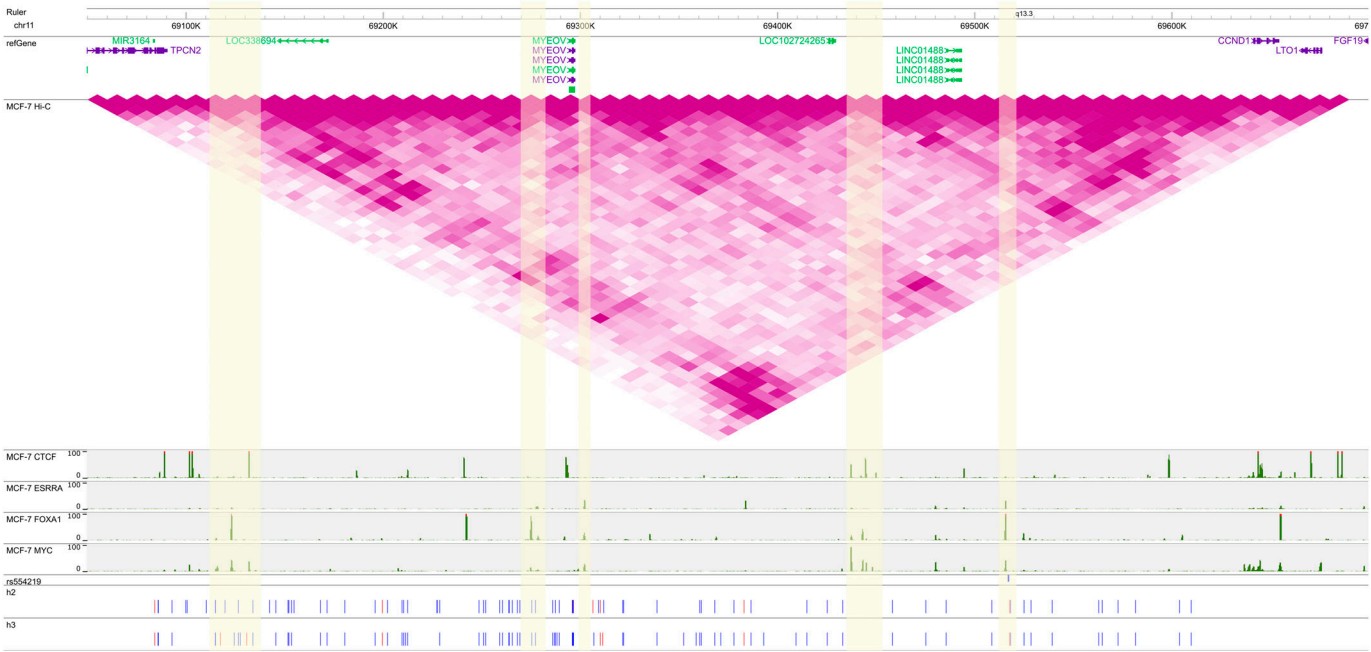

**Figure 1. Rare haplotypes, three-dimensional chromosome organization, and transcription factor-binding at the 11q13 locus.**
ENCODE Hi-C and ChIP-seq (negative log-10 $P$-value) data for CTCF, ESRRA, FOXA1, and MYC are from MCF-7 breast carcinoma cells. VC square root-normalized Hi-C data are plotted on the same intensity scale in all figures. The SNPs of h2 and h3 are indicated below, with blue denoting the reference and red the alternate allele. The GWAS hit rs554219 is indicated in blue. This figure may be viewed online at https://tinyurl.com/yc54zcav.

locus starting from 246 with $P < 1.0 \times 10^{-16}$, each within five iterations. Surprisingly, even though we started at the chr16 locus with the greater number of filtered meta-chromosomes, we found there the lesser number of closed-patterns, suggesting a complex combinatorial scheme underlying the haplotype diversity at these GWAS-hit loci and making it difficult to formulate an a priori rule for knowing the $P$-value threshold which should prevent a combinatorial explosion.

Among the 585,591 + 2,343 and 269,993 + 2,858 noncontiguous closed and original contiguous patterns at the chr10 and chr16 loci, respectively, we evaluated the top 30,019 (with Fisher's exact test $P < 1.0 \times 10^{-50}$) and 29,078 ($P < 1.0 \times 10^{-30}$) in a Cox proportional hazards model for association with breast cancer incidence rates. As at the chr11 locus, none of the original contiguous patterns at these two loci (minimum $P$-values $1.6 \times 10^{-5}$ and $3.2 \times 10^{-4}$) met the inclusion threshold. We discovered, in a similar manner to the chr11 analysis, 10- and 26-SNP haplotypes at the chr10 and chr16 loci, respectively, which were each reduced by recursive partitioning to nine and 20 SNP-alleles (Table S1) without alteration of their frequencies or HRs. As shown in Table 2, these haplotypes, each designated h1, were highly statistically significant in UKBB (and also in DRIVE). Also as previously, the associations were on par with those of the GWAS hits, and the LD of the h1 with their corresponding GWAS hits was strong ($r^2 = 0.70$ and 0.87 and $D' = 0.99$ and 1.00 at the chr10 and chr16 loci, respectively). We therefore again concluded that the common haplotypes h1 were tagging largely the same pool of at-risk subjects as were the GWAS hits.

For the Phase 2 analysis, we extracted SNPs in the TADs containing each h1 using the 3D Genome Browser (Wang et al, 2018) to

predict domain boundaries. Although the h1s of chr10 and chr16 were both located in the center of their respective TADs (Figs S2 and S3), the chr10 TAD contained a well-defined Hi-C block in the region containing and immediately 5′ of h1, whereas the entire TAD of the chr16 locus was rather diffused. Thus, we extracted haplotypes in the regions chr10:120,928,564–121,477,406 (158 SNPs), coinciding with the 5′ boundary of the chr10 TAD and the 5′ boundary of the chr10 h1, and chr16:52,073,187–53,053,996 (195 SNPs), defined by the 5′ and 3′ boundaries of the chr16 TAD and the 5′ and 3′ boundaries of chr16 h1, from 7,976 and 4,595 h1-bearing chromosomes of UKBB breast cancer cases, respectively. These regions contained 5,543 and 3,452 unique contiguous haplotypes and 15,353,437 and 5,954,654 unique meta-chromosomes after the initial overlap, respectively (Table 1). Although there were many more SNPs and unique haplotypes at these loci than at the chr11 locus, the method of using the TAD boundaries (on either both sides or just the longer side of h1) to determine the included SNPs was consistently applied.

As at the chr11 locus, we adjusted the h1-conditional Fisher's exact test $P$-value (among h1 carriers) threshold such that the number of filtered meta-chromosomes was no more than ~30 to prevent a combinatorial explosion. From Iteration 1 onward, a total of 731,526 closed-patterns were found at the chr10 locus starting from 30 with h1-conditional Fisher's exact test $P < 2.0 \times 10^{-6}$ (among h1 carriers) and 2,517,921 at the chr16 locus starting from 25 with $P < 1.5 \times 10^{-5}$ (Table 1). As in Phase 1, we observed that most of the patterns were discovered by Iteration 5. However, at the chr16 locus, an additional four iterations were carried out on

a small number of meta-chromosomes that had not appeared until Iteration 5.

Among the 731,526 + 5,543 and 2,517,921 + 3,452 noncontiguous closed and original contiguous patterns at the chr10 and chr16 loci, respectively, we evaluated the top 256 and 876 (with h1-conditional Fisher's exact test $P < 1.0 \times 10^{-4}$ among h1 carriers only) in a Cox proportional hazards model for association with breast cancer incidence rates, after verifying that none of the original 5,543 and 3,452 contiguous haplotypes (minimum h1-conditional $P$-values $2.2 \times 10^{-4}$ and $1.4 \times 10^{-3}$) met the inclusion threshold. We found 92 and 123 risk-increasing closed-patterns at the chr10 and chr16 loci, respectively, with Cox-LRT $P < 1.0 \times 10^{-5}$. However, only 1 of 92 and 1 of 123 had $P < 0.05$ in the DRIVE replication analysis. The false-positive replication rates from the permutation analysis were estimated to be 3% (1 of 29) and 10% (6 of 60), suggesting that the rare haplotypes at the chr10 and chr16 loci were likely false positives.

### Validation of Chromosome Overlap in a region known to contain a rare risk haplotype

To improve the confidence in our chr11 discoveries, we next sought to apply Chromosome Overlap in a region known to contain a rare risk haplotype. Our recent genome-wide rare-haplotype analysis discovered by a sliding-window approach six loci containing rare haplotypes which were associated with increased breast cancer risk in both UKBB and DRIVE (Wang et al, 2023). The lead 50-SNP haplotype (by $P$-value) discovered at chromosome 22q12, called h38, had a frequency of 10.6 per 10,000 chromosomes and an HR of 2.81 ($P = 1.2 \times 10^{-14}$). We sought to validate the Chromosome Overlap procedure by testing if we could not detect and replicate a non-contiguous rare risk haplotype related to h38.

To achieve this goal, we performed a Phase 1 analysis in region spanned by h38 at chr22:27,749,339–27,874,384 (Table 1 and Fig S4). Note that in the present analysis, h38 contained only 49 SNPs after the exclusion of one SNP that turned out to be triallelic (Table S4). In

addition, this region is near (200–300 kb upstream of), but does not contain, rs62237573, a low-frequency (MAF <1%) SNP listed in the NHGRI-EBI GWAS Catalog (Sollis et al, 2023) for breast cancer risk (EFO_0000305) which displayed moderate but not genome-wide significant evidence for association in UKBB (Table 4).

Table 1 shows that there were 4,221 unique contiguous haplotypes at the chr22 locus among the 18,022 chromosomes of UKBB breast cancer cases and 7,759,059 meta-chromosomes resulting from the initial pairwise overlap. After filtering, we found a total of 398,251 closed-patterns within five iterations starting from 127 with Fisher's exact test $P < 2.0 \times 10^{-12}$. Among the 398,251 noncontiguous closed and 4,221 original contiguous patterns, we evaluated the top 31,858 (with Fisher's exact test $P < 1.0 \times 10^{-9}$) in a Cox proportional hazards model for association with breast cancer incidence rates (see the Materials and Methods section), after verifying that the only original contiguous similar pattern which passed the inclusion threshold was h38. We found that all included patterns had Cox-LRT $P < 1.0 \times 10^{-5}$ in the UKBB discovery analysis, and that 29,081 replicated in DRIVE with $P < 0.05$; however, only a single 34-SNP haplotype remained independently associated after stepwise forward selection. This haplotype, hence reduced to 22 SNPs by recursive portioning without alternation of its haplotype frequency or HR and subsequently designated h1 (Table S4), was both rare (18.5 copies per 10,000 chromosomes in controls) and highly statistically significant in UKBB (HR = 2.58, $P = 8.6 \times 10^{-15}$); in DRIVE, h1 was less strongly associated (OR = 1.39, $P = 4.6 \times 10^{-3}$), but still passed the replication threshold. In comparison, h38 was somewhat rarer (10.8 copies per 10,000 chromosomes) and had a larger but less statistically significant effect in UKBB (HR = 2.89, $P = 4.8 \times 10^{-12}$); in DRIVE; h38's effect was similarly moderated (OR = 1.92, $P = 1.3 \times 10^{-5}$). As h1 was already quite rare and risk-increasing, we did not perform a Phase 2 analysis of this region.

These results suggested that Chromosome Overlap can pick up rare haplotypes at loci known to harbor validated haplotype associations, lending support to the chr11 results in the manner of a positive control. However, as we did not discover h1 in a region

**Table 4. Rare-haplotypes at 22q12, discovery and replication.**

| | HR/OR[a] | Cases chromosomes (per 10,000) | Controls chromosomes (per 10,000) | $P$[b] |
|---|---|---|---|---|
| | | 22q12:27,749,339–27,874,384 | | |
| Variable | | *UKBB discovery* | | |
| h1 | 2.58 | 90 (49.9) | 636 (18.5) | $8.6 \times 10^{-15}$ ($4.1 \times 10^{-15}$) |
| h38 | 2.89 | 59 (32.7) | 371 (10.8) | $4.8 \times 10^{-12}$ ($2.9 \times 10^{-12}$) |
| rs62237573[T] | 1.35 | 233 (129) | 3,318 (96.4) | $1.7 \times 10^{-5}$ ($2.7 \times 10^{-5}$) |
| h1 + rs62237573[T] | 3.07 | 62 (34.4) | 362 (10.5) | $9.9 \times 10^{-14}$ ($4.8 \times 10^{-14}$) |
| | | *DRIVE replication* | | |
| h1 | 1.39 | 196 (32.6) | 120 (23.7) | $4.6 \times 10^{-3}$ ($6.6 \times 10^{-3}$) |
| h38 | 1.92 | 137 (22.8) | 61 (12.1) | $1.3 \times 10^{-5}$ ($2.3 \times 10^{-5}$) |
| rs62237573[T] | 1.33 | 625 (104) | 392 (77.5) | $9.8 \times 10^{-6}$ ($4.4 \times 10^{-6}$) |
| h1 + rs62237573[T] | 2.02 | 136 (22.6) | 57 (11.3) | $3.9 \times 10^{-6}$ ($6.1 \times 10^{-6}$) |

[a]HR (hazard ratio) for UKBB, OR (odds ratio) for DRIVE.
[b]LRT (likelihood ratio test) $P$-value adjusted for age and 10 principal components of ancestry (Fisher's exact test $P$-value).

surrounding and in LD with a GWAS hit, we sought to explain the relationship between h1 and h38 instead. h1 and h38 were in moderate LD with each other ($r^2$ = 0.59, $D'$ = 1.0) and in distinct LD with the GWAS hit, with the $D'$ ($r^2$) of rs62237573 with h1 and h38 being 0.58 (0.068) and 0.90 (0.097), respectively. These values indicate that most chromosomes with h38 also carry the GWAS hit, whereas only a subset of chromosomes with h1 do; thus, h38—but not h1—is like the rare haplotypes on chr11 in respect of its being a subtype of a haplotype bearing the GWAS hit. To find the proportion of the h1 risk attributable to its being a subtype of the GWAS hit, we adjoined rs62237573[T] to the haplotype and reevaluated it in the model. This procedure reduced the cases frequency of h1 (from 49.9 to 34.4 per 10,000) to a value similar to that of h38's (32.7 per 10,000) and increased the LD between the two haplotypes to $r^2$ = 0.82 ($D'$ = 0.91). We drew two conclusions from this observation. First, because h1 + rs62237573[T] and h38 were tagging a highly overlapping set of chromosomes, the additional risk (HR = 3.07 versus 2.58) over carriers of h1 to carriers of h1 + rs62237573[T] was attributable to the contiguous haplotype h38, so that the noncontiguous h1 + rs62237573[T] was therefore a tag for h38. But although it decreased the HR slightly, the removal of rs62237573[T] led to a 45% increase in the size of the h38 at-risk pool. Therefore, we also concluded that the noncontiguous h1 discovered a subset of high-risk individuals not having the GWAS hit.

Finally, we plotted the locations of the SNPs of h1 and h38 to see if they correlated with any ChIP and Hi-C features (Fig S4). The only evident peaks colocalizing with the haplotypes were those of CTCF, and the window did not contain an especially strong Hi-C block. FOXA1-binding occurred in a region devoid of any UKBB-typed SNPs, but without any coincident peaks belonging to other factors. A more intense Hi-C block occurred downstream of h1 and h38 in a region spanned by the original 250-SNP haplotype (Wang et al, 2023). The comparative lack of features about h1 and h38 suggested that the binding events at the chr11 locus may have been specific to regulation of *CCND1* and that some other biological mechanism would have to be sought at the chr22 locus, as explored in our previous study (Wang et al, 2023).

## Discussion

Our findings with Chromosome Overlap support a model wherein rare haplotypes or mutations underlie the germline genetic risk for breast cancer associated with at least some GWAS hits. Despite a genetically heterogeneous replication population, we replicated three noncontiguous rare haplotypes at 11q13 and 22q12 composed of common genotyped SNPs that elevated the risk for breast cancer by 2.6–17-fold. Effects of this size are typically not detected in association analysis. The discoveries at 11q13 and 22q12 represent two distinct mechanisms whereby rare haplotypes can refine the signal tagged by GWAS hits. At the former locus, we found that h2 and h3 were rare subtypes of chromosomes bearing rs554219[G], whereas at the latter, we found that h1 tagged additional risk not captured by carriers of the GWAS hit rs62237573[T]. These two examples illustrate that GWAS hits may label too many or too few individuals as high risk.

We could not replicate the haplotypes discovered at chr10 and chr16 where mechanisms have been proposed to explain the direct effect of alternate alleles of the GWAS hits (Meyer et al, 2008; Cowper-Sal lari et al, 2012). Although we found individual rare haplotypes at each locus, we could not reject the possibility that they replicated in DRIVE solely by chance. These regions therefore may not harbor any rare risk haplotypes. But it is important to keep in mind the potential for imputation and phasing errors when combining individuals across a range of genetic ancestries; to replicate rare haplotypes, not only must the imputation of non-genotyped UKBB variants in DRIVE be accurate, but so must the computational phasing. We attempted to mitigate this issue by phasing and imputing the UKBB and DRIVE genotype data using the TOPMed panel as a common reference, but without whole-genome sequencing, our replication framework had to be based on the strong assumption that both computational steps were accurate. This is a major limitation of our work.

Supporting our replication of h2 and h3 at chr11 is the report of a non-*BRCA1/2* family from the Netherlands with six cases of breast cancer having a strong linkage peak at the 11q13 locus (Rosa-Rosa et al, 2009b). If h2 and h3 are genuine, they could represent a link between a signal identified by two complementary methods, viz., linkage analysis and association studies. The overlap of signals from linkage and association analyses has been demonstrated to be low in the context of type 2 diabetes (McCarthy et al, 2008; Prokopenko et al, 2009), for example, and the reason is thought to be because of the different ages of the causal mutations identified by the two approaches (Ott et al, 2015). According to the theory, when mutations are relatively young, they are in strong linkage disequilibrium with the haplotype on which they arose, and are hence detectable by an excess of haplotype-sharing among affected family members. Over time, recombination breaks the linkage, and the mutation becomes associated with common alleles that are segregating in the population. This mechanism could explain why the frequency spectrum of GWAS hits is so broad (Wray et al, 2011), in contrast to the low-frequency bias predicted under the synthetic-associations model (Dickson et al, 2010). One interpretation of our results, then, is that the rare haplotypes are corrupted versions of an ancestral haplotype on which a single causal variant arose relatively recently. That we could not successfully replicate the chr10 and chr16 haplotypes could simply be a reflection of the fact that the rare variants there arose further in the past. Future applications of chromosome overlap would then have to be focused on linkage peaks and not GWAS hits.

But this model fails to account for the lack of rare variants discovered so far in the vicinity of breast cancer-risk GWAS hits (Lindström et al, 2016; Li et al, 2018) and the comparatively few mutations identified by linkage analysis and subsequent positional cloning (Gonzalez-Neira et al, 2007; Rosa-Rosa et al, 2009a). An alternative hypothesis is that the haplotype itself is the rare causal variant and that risk is because of the interaction of alleles at discrete locations along the haplotype. Two lines of evidence support this hypothesis. The first is the observation that the most statistically significant haplotypes were all noncontiguous, that is, not merely tags of the original contiguous haplotypes. Although the distinction between contiguous and noncontiguous is somewhat arbitrary in our use of only genotyped SNPs, the evidence of a decreasing number of

alleles required for the manifestation of the statistical effect suggests that only a few alleles may be involved in mediating risk. In future studies with a greater number of meta-chromosomes retained after filtering, it should, in principle, be possible to reduce the haplotype SNP-alleles down to only the most essential set.

The second line of evidence in support of the noncontiguous rare-haplotype hypothesis is the ChIP and Hi-C data in Fig 1 showing the potential for biologically relevant interactions between key SNPs of h2 and h3. For example, the chromosomal loops on chr11 are complemented by the binding of CTCF, FOXA1, and ESRRA, molecules with potential roles in breast cancer risk. Regarding ESRRA, it is known that the closely related estrogen receptor (ER) binds throughout the genome at enhancers distant from the start sites of genes it regulates (Carroll et al, 2005; Eeckhoute et al, 2006), including *CCND1*. Several proteins are involved in ER recruitment, including the pioneer factor FOXA1 (Eeckhoute et al, 2006; Hurtado et al, 2011) and CTCF, which acts upstream of FOXA1 to drive ER-mediated transcription via chromosome loops (Zhang et al, 2010) and partitions the genome into ER-responsive blocks (Chan & Song, 2008). A small fraction of binding events (e.g., less than 5% of those of CTCF in MCF-7 cells) involve all three factors, which then contribute to the down-regulation of estrogen target genes (Hurtado et al, 2011; Ross-Innes et al, 2011). A fourth TF with frequent binding in the *CCND1* TAD is encoded by the proto-oncogene *MYC*. MYC-binding was identified by computational analysis in a *CCND1* enhancer encompassing the original GWAS hit rs614367 (Wang et al, 2019), later replaced with rs554219 using fine-mapping (French et al, 2013), and it has long been known that MYC can repress *CCND1* expression (Jansen-Dürr et al, 1993; Philipp et al, 1994). In our data, the haplotype SNP-alleles of h2 and h3, especially when in the alternate phase, often colocalized with ChIP peaks, suggesting that chromosome overlap is detecting haplotypes involved in *CCND1* regulation. Without corroborating experimental evidence, however, our statistical results cannot be used to definitively distinguish between a rare haplotype and a rare-variant model of breast cancer risk.

Nevertheless, our use of closed-pattern mining has made it possible to find rare, risk haplotypes and generate hypotheses that bear directly on the biological consequences of GWAS hits, and it is important to assess the advantages and limitations of this method in future genetic association studies. The principal advantage of closed-pattern mining over other pattern-mining algorithms is its elimination of redundant patterns which occur on one and the same sets of chromosomes. In this regard, Chromosome Overlap is similar to another closed-pattern miner called LCM (Uno et al, 2004) which has been used in a number of recent studies to detect combinations of transcription factor-binding events (Terada et al, 2013), expressed mRNAs (Relator et al, 2018), and SNP genotypes (Terada et al, 2016; Yoshizoe et al, 2018). But where LCM is a "bottom-up" method that extends shorter closed-patterns to longer ones, Chromosome Overlap is a "top-down" method similar to an older algorithm called CARPENTER (Pan et al, 2003) which finds long patterns by overlap. And although use of LCM and Chromosome Overlap both require pruning of the number of SNPs in some fashion (Relator et al, 2018; Yoshizoe et al, 2018), Chromosome Overlap is suited to finding the long, rare patterns that LCM would find only at the end of its search. A potential limitation of our method, however, is that the closed patterns which Chromosome Overlap discovers among cases are not necessarily closed patterns among controls, or in an independent replication population; at best, we can compare the candidate closed-pattern in one population to its *closure* in another. This is not so much a problem as it is a refinement of the statistical hypothesis, namely, that candidate patterns discovered in cases should have increased frequency among cases by virtue of their being genuine risk-increasing patterns; risk-decreasing patterns should only be shared by controls, who were too numerous to analyze in our study.

# Materials and Methods

### Chromosome Overlap: overview

Chromosome Overlap consists in iteratively overlapping regions of chromosomes from affected individuals and looking for shared haplotypes that are enriched in cases. First, all pairwise overlaps of the unique chromosomes from $N$ affected individuals are formed to find shared, chiefly noncontiguous, closed haplotype patterns called *meta-chromosomes* (in this article, "pattern," "haplotype," and "meta-chromosome" are used synonymously). Most of these patterns are filtered out, and the remainder are retained for additional rounds of pairwise overlap. The process of forming all pairwise overlaps of a set of meta-chromosomes is known as an *iteration*, numbered from 0 to count the total number of times the process has been performed, and a pattern is said to have been discovered at the first iteration in which it appears as the longest pattern shared by a pair of meta-chromosomes. Closed haplotype patterns which have been discovered at previous iterations are not included in the next round of overlaps, and the process ceases when no more new patterns are discovered. A mathematical justification that this procedure discovers all closed-patterns among—or what is the same thing, forms all possible groupings of—the filtered set of meta-chromosomes is given first, and a scheme that allows the overlaps to be computed efficiently in parallel is described second. Following is a description of how Chromosome Overlap was applied in this study, including data sources, computational considerations, and statistical analyses.

### Chromosome Overlap: mathematical formulation

If the $i^{th}$ chromosome in a sample is defined by the vector $\mathbf{x}_i = (x_{i,1},...,x_{i,m})$ of alleles $x_{i,k} = s_k$ at $m$ biallelic SNPs, then a *pattern* is a list $h = (k_1, s_{k_1}), (k_2, s_{k_2}),...,(k_l, s_{k_l})$ of $l \leq m$ alleles. A chromosome $\mathbf{x}_i$ is said to contain a pattern $h$ if $x_{i,k_j} = s_{k_j}$ for all $j \in \{1,...,l\}$, and a group of chromosomes is said to share a pattern which is contained by each of the $\mathbf{x}_i$. A pattern is said to be *closed* if it is the longest pattern shared by a group of chromosomes, or what is the same thing, if it is the intersection or overlap of a group of chromosomes.

Overlap of a number $\sigma$ of chromosomes produces a pattern which is shared by all $\sigma$, that is, the set $h = \{(k_j, s_{k_j}) | x_{i_1,k_j} = \cdots = x_{i_\sigma,k_j} = s_{k_j}\}$; such a pattern is called a *meta-chromosome*. A meta-chromosome is said to comprise not the chromosomes which contain it, but only those particular chromosomes overlapped to form it, so that $h = g$ may be two equivalent patterns which comprise different sets of chromosomes.

The overlap of a number $\sigma$ of meta-chromosomes produces a *meta-meta-chromosome* defined by the intersection $h' = h_{i_1} \cap \cdots \cap h_{i_\sigma}$. If each meta-chromosome $h_i$ should comprise exactly $\sigma > 1$ distinct chromosomes, then $h'$ may comprise as few as $\sigma + 1$ chromosomes—for example, if any $h_{i_j}$ should share all but one chromosome with any other $h_{i_{j'}}$, then $\sigma = 3$-tuples, say, comprising 123, 124, and 134 will produce the $\sigma+1$-tuple 1234. Thus, it is always possible to add exactly one of each unique chromosome to any meta-chromosome. This feature is known as the *add-one property* and can be used to show that iteration of chromosome overlap forms all possible chromosome combinations, not just a sequence of $\sigma$-tuples, $\sigma^2$-tuples, etc. For if the process should be iterated such that all $\sigma$-overlaps of meta-chromosomes $h^{(r-1)}$ comprising $\sigma+r-1$ chromosomes are formed by the start of iteration $r \geq 1$, then all meta–meta-chromosomes comprising $\sigma+r$ chromosomes will be produced by the start of iteration $r+1$. Hence, iterated overlap generates all closed-patterns shared by at least $\sigma$ chromosomes. If the meta-chromosomes should be filtered at some iteration (e.g., $r = 1$ in the main text) such that only a subset are kept for further overlaps, then the closed-patterns found become only the closed-patterns of the filtered set, and the word "chromosome" should be replaced with "filtered meta-chromosome."

Once all overlaps are computed at iteration $r$, the unique meta-chromosomes are found by pruning the list of duplicates. Now, if a meta-chromosome $h^{(r'-1)}$ appearing at iteration $r'$ should appear again at iteration $r>r'$, then $h^{(r'-1)}$ actually will be found to comprise (at least) $\sigma+r-1$ chromosomes. But as the previous iterations have already added every single chromosome one-by-one to $h^{(r'-1)}$ via the add-one property, it will not necessary be to overlap $h^{(r-1)} = h^{(r'-1)}$ in iteration $r$ in which $\sigma+r$-tuples are generated from $\sigma+r-1$-tuples which each share all but one chromosome with $h^{(r-1)}$. Thus, before the start of iteration $r+1$, all meta–meta-chromosomes which are found to have been generated in previous iterations are purged from the list. The process is complete at the first iteration in which no patterns are found to be novel.

### Triangular-array algorithm

Here, we describe the algorithm for determining the chromosome combination corresponding to an arbitrary index. Let $N$ be the number of samples and $\sigma$ be the degree of overlap. Our problem is to find, for a given $N$ and $\sigma$, what chromosome combination is the $I^{\text{th}}$? To answer this question without generating a list of all $\binom{2N}{\sigma}$ chromosome combinations, we first index each combination by $I \in \left\{ 0, 1, \ldots, \binom{2N}{\sigma} - 1 \right\}$ corresponding to a unique multiindex $\mathcal{I} = i_1, i_2, \ldots, i_\sigma$ of $\sigma$ different chromosomes with $i_1 < i_2 < \cdots < i_\sigma$. The mapping from $I$ to $\mathcal{I}$ is accomplished iteratively using the following lemma. In what follows, we put $N \leftarrow 2N$ for notational convenience.

### Lemma 1.

For any integers $N$ and $\sigma \leq N$, $\binom{N}{\sigma} + \binom{N-1}{\sigma} + \cdots + \binom{N-(N-\sigma)}{\sigma} = \binom{N+1}{\sigma+1}$.

*Proof.* We will prove the lemma by induction. Clearly the result holds when $\sigma = 1$ as the usual integer-summation formula, and the result when $\sigma = N$ is trivial. Now, suppose that each term on the l.h.s. is the number of ways to arrange $N-j$ things ($0 \leq j \leq N-\sigma$) in a $\sigma$-dimensional triangular array (Fig S5), thus, $\binom{N}{\sigma}$ corresponds to the number of arrangements of the items $\{2, \ldots, N+1\}$, $\binom{\sigma}{\sigma}$ to the arrangements of $\{N+1\}$, and, in general, $\binom{N-j}{\sigma}$ to arrangements of $\{j+2, \ldots, N+1\}$. Now, append to each arrangement an element $i_1$ which is one less than the minimum item $j+2$ of each array. In this way, we form all $\sigma$-combinations of the elements $\{1, 2, \ldots, N+1\}$, of which there are $\binom{N+1}{\sigma+1}$. Because the l.h.s. is equal to the r.h.s., the result follows for all $\sigma$ up to $N$.

To use Lemma 1, arrange the values of $\mathcal{I}$ in a $\sigma$-dimensional triangular array such that $i_1$ takes values in $\{1, \ldots, N-\sigma+1\}$, $i_2$ takes values in $\{i_1+1, \ldots, N-\sigma+2\}$, and in general, $i_j$ takes values in $\{i_{j-1}+1, \ldots, N-\sigma+j\}$ (Fig S5). After the convention that the first dimension of a matrix is its rows, we have that $i_1$ corresponds to the $\sigma^{\text{th}}$ dimension of a matrix is its rows, we have that $i_1$ corresponds to the $\sigma^{\text{th}}$ dimension, $i_\sigma$ the first, and $i_j$ the $\sigma-j+1^{\text{th}}$. Starting from $i_1 = 1$, $i_2 = 2, \ldots, i_\sigma = \sigma$ and incrementing from right to left (i.e., ranging through all values in the first dimension before changing the value in the second), the $I^{\text{th}}$ element will be the multiindex $\mathcal{I}$. Determining the indices $i_j$ is equivalent to determining the value or *layer* of the $j^{\text{th}}$ dimension of the array in which the $I = I^{(0)\text{th}}$ element resides. To determine the first layer, recognize that, for fixed $i_1$, there are only $\binom{N-i_1}{\sigma-1}$ chromosome combinations available to the remaining indices, because $\forall j > 1$, $i_j > i_1$. Then, $i_1$ is determined by subtracting layers in the $\sigma^{\text{th}}$ dimension until the smallest $k$ is found which satisfies $I^{(0)} - \sum_{l=0}^{k} \binom{N-l}{\sigma-1} < 0$, or equivalently (by Lemma 1) $\varepsilon_1 := \left[ \binom{N}{\sigma} - \binom{N-k}{\sigma} \right] - I^{(0)} > 0$, and putting $i_1 \leftarrow k$. The error $\varepsilon_1$ is the remainder beyond $I^{(0)}-1$ in the $\sigma-1$-dimensional array, which has $\binom{N-i_1}{\sigma-1}$ combinations. Thus, after determining $i_j$, put $I^{(j)} \leftarrow \binom{N-i_j}{\sigma-j} - \varepsilon_j$ and get the position of the multiindex $i_{j+1}, \ldots, i_\sigma$ in the $\sigma-j$-dimensional array of $\binom{N-i_j}{\sigma-j}$ combinations. To do so, solve again for the smallest $k$ such that $\varepsilon_{j+1} := \left[ \binom{N-i_j}{\sigma-j} - \binom{N-i_j-k}{\sigma-j} \right] - I^{(j)} > 0$ and put $i_{j+1} \leftarrow i_j+k$, because $i_{j+1}$ starts from $i_j+1$. The algorithm terminates when $I^{(\sigma-1)} = \binom{N-i_{\sigma-1}}{1} - \varepsilon_{\sigma-1}$ is the position of the multiindex $i_\sigma$ in the one-dimensional array of $N-i_{\sigma-1}$ "combinations" of single chromosomes.

The foregoing code is implemented in the R script index2-combo2.R which takes parameters I (the position), n (the total number of chromosomes), and sigma (the degree of overlap). We use zero-based indexing so that I ranges from 0 to $\binom{n}{\sigma} - 1$. The output is a list $i_1, \ldots, i_\sigma$ of strictly increasing numbers corresponding to the multiindex $\mathcal{I}$. If instead a non-decreasing sequence is desired, the parameter allow.repeats can be used which increases the total number of chromosomes by $\sigma-1$. With this modification, each $i_j$ beyond $i_1$ can begin at exactly $i_{j-1}$ if combinations are subject to the replacement $i_j \leftarrow i_j-j+1$.

## Filtering

In the initial pairwise overlap of the unique chromosomes from $N$ individuals, there are a maximum of $\binom{2N}{2}$ distinct meta-chromosomes, which number will quickly grow to an astronomical size upon iteration. To limit the growth rate, a strict $P$-value threshold was implemented to select only those haplotype patterns most associated with breast cancer risk in the discovery dataset (see below), and subsequent overlap iterations were performed only on this set. The threshold was set for each examined locus in such a way that the maximum number of meta-chromosomes generated in any iteration was between $10^5$ and $10^6$. The less stringent the threshold, the lesser was our chance of missing a risk haplotype appearing for the first time at a later iteration.

An additional filtering step was applied when the primary filtering was not enough to dampen the growth rate: when a family of patterns not related as subset–superset had the same frequency and crude breast cancer association OR, only the member with the fewest alleles was carried forward. Although not theoretically justified, we reasoned that this filtering removed long patterns that differed by at most a few alleles.

## Computational considerations

Chromosome overlap is implemented in the IBM high-performance–computing environment LSF and is generalizable to any degree of overlap $\sigma$ using an algorithm that iteratively and recursively finds which tuple $i_1, i_2,..., i_\sigma$ in a $\sigma$-dimensional triangular array (Fig S5) corresponding to the $l^{\text{th}}$ overlap of $\sigma$ meta-chromosomes (see above), alleviating the need to generate the complete list of $\binom{N_r}{\sigma}$ overlaps of $N_r$ meta-chromosomes at each iteration $r \geq 0$ and allowing the operations to be computed in parallel. In this article, we only consider pairwise overlap, that is, $\sigma = 2$. This restriction means that we potentially filter out risk-associated patterns which appear for the first time when three or more chromosomes are overlapped, but ensures that we test all patterns shared by at least two.

## Haplotype data

Publicly available haplotype data from the UKBB was used for the discovery analysis. As described previously (Bycroft et al, 2018), this SHAPEIT3-phased dataset consists of 487,409 samples phased at 658,720 autosomal SNPs on the UKBB Axiom Array. The discovery analysis included a total of 181,034 women from the UKBB "white British" ancestry subset after excluding those who (1) were identified to be outliers in heterozygosity or genotype missingness rates; (2) showed any sex chromosome aneuploidies; (3) were second-degree or closer relatives of any or third-degree relatives of more than 10 other genotyped individuals or (4) withdrew from UKBB before this study began. Genotype principal components for the discovery analysis in the 181,034 women were obtained from the UKBB portal. The preceding steps were performed in PLINK (Purcell et al, 2007) and KING (Manichaikul et al, 2010), as described in our

previous study (Wang et al, 2023). Breast cancer (UKBB data-field 40006, ICD10 code C50) was reported in 9,011 women with a mean (SD) age of onset of 56.2 (8.6) years; the remaining 172,023 women free of breast cancer had a mean (SD) age of 65.0 (7.9) years at the end of follow-up.

All of the UKBB genotype and phenotype data used in this study are available from the UKBB Portal (https://www.ukbiobank.ac.uk/).

Female subjects in the Discovery, Biology, and Risk of Inherited Variants in Breast Cancer (DRIVE) study were used for the replication analysis. Briefly, the DRIVE study was initiated in 2010 as part of the NCI's Genetic Associations and Mechanisms in Oncology initiative (http://epi.grants.cancer.gov/gameon/) and includes data from 60,015 breast cancer cases and controls genotyped on the custom Illumina OncoArray (Amos et al, 2017). The 528,620 OncoArray SNPs were filtered to remove SNPs which (1) had genotype missingness rate >10%, (2) were monomorphic or (3) were not in Hardy–Weinberg equilibrium ($P < 1 \times 10^{-10}$), leaving a total of 433,297. A total of 4,669 subjects were excluded who (1) showed any sex chromosome aneuploidies; (2) had genotyping rates <90%; (3) were identified as male by PLINK's sex check; (4) were second degree or closer relatives of any other subject; (5) were not of European ancestry according to principal components analysis including the 1,000 Genomes Project Phase 3 data or (6) had missing age data, all as described in our previous study (Wang et al, 2023). The preceding steps were performed in PLINK (Purcell et al, 2007), KING (Manichaikul et al, 2010), and FlashPCA2 (Abraham et al, 2017), leaving 30,064 cases (mean [SD] age at onset: 61.7 [10.7] years) and 25,282 controls (mean [SD] age at end of follow-up: 59.6 [10.7] years).

All of the DRIVE data used in this study are publicly available from dbGaP under accession number phs001265.v1.p1.

Although UKBB contained fewer breast cancer cases than did DRIVE, the former dataset was chosen for the discovery analysis because of its greater genetic homogeneity. To improve the coverage of UKBB Axiom SNPs on the DRIVE OncoArray and minimize systematic phasing errors between the datasets, we carried out genotype imputation as part of our previously published genome-wide rare-haplotype analysis (Wang et al, 2023). Genotype imputation and phasing were performed on the TOPMed Imputation Server (https://imputation.biodatacatalyst.nhlbi.nih.gov) running Minimac4 (Das et al, 2016) and Eagle2 (Loh et al, 2016), with the TOPMed panel including 194,512 haplotypes and 308 million sequenced variants being used as the reference (https://topmed.nhlbi.nih.gov) (Taliun et al, 2021). Only high-quality imputed SNPs (with Minimac Rsq ≥0.8) that were genotyped on the UKBB Axiom Array and biallelic in both cohorts (after eliminating alleles with MAF = 0) were used to form haplotypes in the selected regions. All genomic coordinates quoted in this article are with respect to the GRCh38 *Homo sapiens* assembly.

## ENCODE data

ChIP-seq and HiC data referenced in this study are available from the ENCODE Portal (https://www.encodeproject.org/) with accession numbers ENCSR000DML (MCF-7 CTCF, *P*-value bigWig file: ENCFF877ZYR), ENCSR954WVZ (MCF-7 ESRRA, *P*-value bigWig file: ENCFF059ZSC), ENCSR126YEB (MCF-7 FOXA1, *P*-value bigWig file: ENCFF512UGW), ENCSR000DMQ (MCF-7 MYC, *P*-value bigWig file:

ENCFF149FXH), and ENCSR660LPJ (MCF-7 Hi-C, contact matrix file: ENCFF420JTA).

## Statistical analysis

All statistical analyses described in this article were carried out in R (R Core Team, 2022) version 3.6.1.

## Two-phase analysis

Chromosome Overlap was applied in two phases on the case-only portion of the UKBB discovery dataset. In Phase 1, a common haplotype linked to the GWAS hit was discovered. In Phase 2, rare subtypes of the common haplotype were discovered via conditional analysis.

### *Phase 1: common-haplotype discovery*

For Phase 1, haplotypes containing UKBB-genotyped SNPs in a region ±100 kb from a GWAS hit were extracted using BCFtools (Li, 2011) version 1.10.2; if the number of genotyped SNPs was greater than 60, the region was reduced symmetrically. After forming all pairwise overlaps of chromosomes from affected individuals, the association of each meta-chromosome with breast cancer was assessed using Fisher's exact test in all UKBB breast cancer cases and controls. The top ~100 patterns in Iteration 1 by *P*-value were retained for iterated overlap to obtain the complete set of closed-patterns, whose associations were assessed by Fisher's exact test upon completion of the last iteration. The top ~20–30,000 haplotypes by *P*-value were then further evaluated one-by-one for their association with breast cancer incidence rates under an additive genetic model using a Cox proportional hazards model having age as the time axis. All analyses were adjusted for the first 10 genotype principal components, and the number (zero, one, or two) of copies of each haplotype was modelled as a continuous variable. The statistical significance of the adjusted hazard ratio (HR) of each haplotype was evaluated using the one-degree-of-freedom LRT. The single most significant haplotype was designated h1 after verifying by stepwise forward-selection that no other haplotype had $P < 1.0 \times 10^{-5}$ relative to the model with h1.

### *Phase 2: rare-haplotype discovery*

In Phase 2, h1-bearing case chromosomes in an extended region covering ~100–200 SNPs in a ~1 Mb region were extracted using BCFtools (Li, 2011) version 1.10.2. The region was chosen based on the boundaries of the TAD defined by the 3D Genome Browser (Wang et al, 2018) (Figs S1–S3) and the location of h1 so as to include one or both sides of the TAD between h1 and the TAD boundary. After forming all pairwise overlaps of h1-bearing chromosomes from breast cancer cases, the association of each meta-chromosome with breast cancer incidence rates over and above the risk attributable to h1 was assessed using an "h1-conditional" Fisher's exact test, that is, one restricted to h1-bearing chromosomes only. The top ~30 patterns in Iteration 1 by *P*-value were retained for iterated overlap to obtain the complete set of closed-patterns, whose associations were

assessed by Fisher's exact test upon completion of the last iteration. The resulting haplotypes with Fisher's exact test $P < 1.0 \times 10^{-4}$ were further evaluated one-by-one for their association with breast cancer incidence rates under an additive genetic model using a Cox proportional hazards model having age as the time axis. All analyses were adjusted for h1 and the first 10 genotype principal components, and the number (zero, one, or two) of copies of each haplotype was modelled as a continuous variable. The statistical significance of the adjusted HR was evaluated using the one-degree-of-freedom LRT relative to the model with h1 alone. Haplotypes were carried forward for replication which had $P < 1.0 \times 10^{-5}$ relative to the model with h1 alone.

## Replication

The association of h1 and each of its rare subtypes with breast cancer risk was assessed in DRIVE under an additive genetic model using logistic regression, the method appropriate to case-control designs. All analyses were adjusted for age and the first 10 genotype principal components, and the number (zero, one, or two) of copies of each haplotype was modelled as a continuous variable. The statistical significance of the OR was evaluated for each discovered haplotype in Phase 2 individually using the one-degree-of-freedom LRT relative to the model with h1 alone; a similar confirmation was done in Phase 1 relative to the null model. Successful replication was declared at *P* < 0.05.

## Final set of rare risk haplotypes

The haplotypes which replicated in DRIVE (excepting h1) were again assessed in UKBB using stepwise forward selection. Haplotypes were sequentially added to the model which had LRT $P < 1.0 \times 10^{-5}$ relative to the previous model, beginning with the model having h1 alone. Those haplotypes surviving selection were designated h2, h3, etc., in order of selection; all were rare subtypes of h1.

## Permutation analysis

The empirical false-positive rate of rare-haplotype replication was determined by permuting the case/control labels of h1-carriers in such a way that the number of h1 hetero- and homozygotes was preserved, and then repeating Chromosome Overlap on chromosomes from permuted pseudo-cases. In this way, the h1 association with breast cancer incidence rates from Phase 1 had been preserved, and the expected number of rare haplotypes that replicated due merely to chance (i.e., the false-positive discovery rate) was estimated by computing the fraction of discovered haplotypes with $P < 1.0 \times 10^{-5}$ and HR > 1 in the permuted UKBB dataset which replicated in DRIVE with *P* < 0.05 and OR > 1.

## Haplotype reduction

The minimal set of alleles necessary to define each h1 in Phase 1 was determined using rpart (Therneau & Atkinson, 1997), the recursive partitioning procedure implemented in R, with parameters cp = −1 and minsplit = 1 to ensure that the full tree was grown within

a maximum of 30 steps. Alleles for splitting at each step were chosen which maximized the Gini impurity reduction. Recursive partitioning was not used for rare haplotypes in Phase 2 where the small number of haplotype carriers made it difficult to define a minimal set of alleles.

## Data Availability

The code generated during this study is available on GitHub at https://github.com/wletsou/ChromosomeOverlap.

## Supplementary Information

## Acknowledgements

The authors thank the high-performance–computing facility at St. Jude Children's Research Hospital for computational support. This work was supported by R01CA216354 from the US National Cancer Institute (W Letsou, W Moon, C Im, Y Sapkota, LL Robison, and Y Yasui), T32CA225590 from the US National Cancer Institute (W Letsou), the American Lebanese Syrian Associated Charities (W Letsou, F Wang, Y Sapkota, LL Robison, & Y Yasui), the Alberta Machine Intelligence Institute (C Im & Y Yasui), and the New York Institute of Technology College of Arts & Sciences (W Letsou).

### Author Contributions

W Letsou: conceptualization, software, formal analysis, methodology, and writing—original draft, review, and editing.
F Wang: software, methodology, and writing—review and editing.
W Moon: software.
C Im: methodology and writing—review and editing.
Y Sapkota: methodology and writing—review and editing.
LL Robison: funding acquisition, methodology, and writing—review and editing.
Y Yasui: conceptualization, formal analysis, supervision, funding acquisition, methodology, and writing—original draft, review, and editing.

### Conflict of Interest Statement

The authors declare that they have no conflict of interest.

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
