## [Reviewer comments · Life Science Alliance]

Life Science Alliance

Refining the genetic risk of breast cancer with rare haplotypes and pattern-mining

William Letsou, Fan Wang, Won Jong Moon, Cindy Im, Yadav Sapkota, Leslie Robison, and Yutaka Yasui

DOI: <https://doi.org/10.26508/lsa.202302183>

Corresponding author(s): William Letsou, New York Institute of Technology and Yutaka Yasui, St. Jude Children's Research Hospital

Review Timeline:

Submission Date:	2023-05-24
Editorial Decision:	2023-06-28
Revision Received:	2023-07-07
Editorial Decision:	2023-07-24
Revision Received:	2023-07-24
Accepted:	2023-07-26

Transaction Report:

June 28, 2023

Re: Life Science Alliance manuscript #LSA-2023-02183-T

William Letsou
New York Institute of Technology

Dear Dr. Letsou,

Thank you for submitting your manuscript entitled "Refining the genetic risk of breast cancer with rare haplotypes and pattern-mining" to Life Science Alliance. The manuscript was assessed by an expert reviewer, whose comments are appended to this letter. We invite you to submit a revised manuscript addressing the Reviewer comments.

When submitting the revision, please include a letter addressing the reviewer comments point by point.

Thank you for this interesting contribution to Life Science Alliance. We are looking forward to receiving your revised manuscript.

Sincerely,

B. MANUSCRIPT ORGANIZATION AND FORMATTING:

Reviewer #1 (Comments to the Authors (Required)):

This manuscript represents a further development of a method the authors have published in 2022. In their previous paper, they were associating haplotypes with disease (Breast cancer), with haplotypes being sequences of alleles at contiguous SNPs. In this new manuscript, they relaxed the requirement that alleles be from contiguous alleles and allow for sets of alleles at any SNPs. This extension brings the authors into the realm of pattern mining as they rightfully point out. I consider this extension a major step forward, not just a minor improvement. The manuscript is well written, but I have some comments as outlined below. The manuscript is also rather long and wordy and I wish it could be shortened quite a bit.

My main point is the authors' reference to closed pattern mining. Is it necessary that their patterns be closed? What if they were not closed, would that make any difference to the authors' methods and conclusions? The authors should carefully defend their reference to the closedness of their patterns. It seems to me they might as well just refer to patterns and omit "closed".

The authors mention in the abstract that their method does not amount to an exhaustive enumeration of allele patterns. In the Introduction on page 4, they mention their "new ... algorithm called Chromosome Overlap" but should add a little more detail. For example, they might specify that they start with some short allele patterns and extend these iteratively in some way, or the other way around. They do present detail on page 18 but a few words at this point would be helpful.

Detailed comments are as follows.

Page 3, line 7 up (from the bottom): "This study identified" presumably refers to the Wang et al. 2023 paper. If so, it should be "That study"; "this study" would refer to their current manuscript.

Page 3, line 2 up: The authors mention that their approach has not previously been considered. True enough, but they should also mention that various publications refer to genotype patterns, with genotypes from noncontiguous SNPs. They should quote, for example, <https://doi.org/10.1016/j.tig.2022.04.009>.

Page 4, line 11: The authors quote various papers as having applied closed pattern mining, but not all of these papers work with closed patterns. For example, the Pounraja and Girirajan 2022 paper works at the level of genes.

Two lines down: The authors write "As it is haplotypes that are passed from parent to offspring" -- I don't understand the argument here. The only relevant fact seems whether alleles or genotypes are disease associated, no matter how they are passed from parent(s) to offspring.

Page 7, line 2: "hit's" should be "hits", or "hit" if it was only a single hit.

Page 7, line 5: "underlay" should be "underlie".

Page 10, line 2 up: "As at the" should probably just be "At the".

Page 14, line 7: "linked to a GWAS hit" seems to refer to genetic linkage. The authors should not use "linkage" and "linked" (there is no genetic linkage here) but instead refer to (genetic) association.

Page 14, line 8: Should "explicate" be "explain"?

Page 18, "Methods" line 3: "consists in" should be "consists of".

Page 19, last line: After all the high-flying theory, the authors mention here that they "only consider $\sigma = 2$ ". They should describe in lay language what this means, and also make this clear early on in the manuscript.

Manuscript # LSA-2023-02183-TR Refining the genetic risk of breast cancer with rare haplotypes and pattern-mining

We would like to thank the editor and reviewer for their time and consideration of our manuscript. Our responses to each of the comments we received are given below, with changes to the manuscript text shown in italic.

Response to Reviewer Comments

This manuscript represents a further development of a method the authors have published in 2022. In their previous paper, they were associating haplotypes with disease (Breast cancer), with haplotypes being sequences of alleles at contiguous SNPs. In this new manuscript, they relaxed the requirement that alleles be from contiguous alleles and allow for sets of alleles at any SNPs. This extension brings the authors into the realm of pattern mining as they rightfully point out. I consider this extension a major step forward, not just a minor improvement. The manuscript is well written, but I have some comments as outlined below. The manuscript is also rather long and wordy and I wish it could be shortened quite a bit.

We thank the reviewer for these positive comments on our approach in this paper being orthogonal to our previous work on contiguous-SNP haplotypes. We have attempted to shorten the manuscript where possible, and hope that our responses have addressed the length concern.

My main point is the authors' reference to closed pattern mining. Is it necessary that their patterns be closed? What if they were not closed, would that make any difference to the authors' methods and conclusions? The authors should carefully defend their reference to the closedness of their patterns. It seems to me they might as well just refer to patterns and omit "closed".

The reviewer is right that not all patterns are closed patterns, and that by virtue of our case-only design in the discovery phase, non-closed patterns can be found which distinguish cases from controls. We therefore have revised the Introduction to clarify that closed patterns are the longest patterns shared by a group of chromosomes, that non-closed patterns are shorter patterns shared by the same set of chromosomes, and that non-closed patterns are therefore redundant with their corresponding closed patterns. We have addressed this comment in the revised paper by modifying the sentence, "Applied to haplotype analysis, a pattern of SNP-alleles is said to be closed if it is the longest pattern shared by a group of chromosomes, *capturing one and the same set of chromosomes as certain shorter patterns and therefore obviating the need for exhaustive enumeration,*" (p 5) and adding a definition of closure: "Applied to market data, a pattern of goods is said to be *closed* if no item can be added to it without diminution of the number of transactions in which the pattern appears, *and the closure of any pattern is the shortest closed pattern which contains it as a subset*" (p 5).

We have also appraised the advantages and limitations of our approach in a revised version of the final paragraph: "*Nevertheless, our use of closed-pattern mining has made it possible to find rare, risk haplotypes and generate hypotheses that bear directly on the biological consequences*

of GWAS hits, and it is important to assess the advantages and limitations of this method in future genetic association studies. The principal advantage of closed-pattern mining over other pattern-mining algorithms is its elimination of redundant patterns which occur on one and the same sets of chromosomes. In this regard, Chromosome Overlap is similar to another closed-pattern miner called LCM (Uno et al, 2004) which has been used in a number of recent studies to detect combinations of transcription factor binding events (Terada et al, 2013), expressed mRNAs (Relator et al, 2018) and SNPs (Terada et al, 2016; Yoshizoe et al, 2018). But where LCM is a “bottom-up” method that extends shorter closed patterns to longer ones, Chromosome Overlap is a “top-down” method similar to an older algorithm called CARPENTER (Pan et al, 2003) which finds long patterns by overlap. And while use of LCM and Chromosome Overlap both require pruning of the number of SNPs in some fashion (Relator et al, 2018; Yoshizoe et al, 2018), Chromosome Overlap is suited to finding the long, rare patterns that LCM would find only at the end of its search. A potential limitation of our method, however, is that the closed patterns which Chromosome Overlap discovers among cases are not necessarily closed patterns among controls or in an independent replication population: at best we can compare the candidate closed pattern in one population to its closure in another. This is not so much a problem as it is a refinement of the statistical hypothesis, namely, that candidate patterns discovered in cases should have increased frequency among cases by virtue of their being genuine risk-increasing patterns; risk-decreasing patterns should only be shared by controls, who were too numerous to analyze in our study.” (p 17).

The authors mention in the abstract that their method does not amount to an exhaustive enumeration of allele patterns. In the Introduction on page 4, they mention their “new ... algorithm called Chromosome Overlap” but should add a little more detail. For example, they might specify that they start with some short allele patterns and extend these iteratively in some way, or the other way around. They do present detail on page 18 but a few words at this point would be helpful.

We thank the reviewer for pointing this out. We agree that the transition from the Introduction to the Results is rather abrupt, and, in response to this comment, have adapted a sentence from the first paragraph of the Methods: “*Our method consists in iteratively overlapping pairs of chromosomes from affected individuals and looking for shared, chiefly non-contiguous haplotypes. We then compare the counts of (the closures of) these patterns in cases and controls on the hypothesis that the sharing of patterns by cases should be associated with their being cases*” (p 5). In addition, we indicate here that it is pairs of chromosomes which are being overlapped, in response to the reviewer’s comment below about “ $\sigma = 2$ ” showing up abruptly in the Methods.

Detailed comments are as follows.

Page 3, line 7 up (from the bottom): “This study identified” presumably refers to the Wang et al. 2023 paper. If so, it should be “That study”; “this study” would refer to their current manuscript.

Thank you, this change has been accepted and incorporated.

Page 3, line 2 up: The authors mention that their approach has not previously been considered. True enough, but they should also mention that various publications refer to genotype patterns, with genotypes from noncontiguous SNPs. They should quote, for example, <https://doi.org/10.1016/j.tig.2022.04.009>.

We appreciate the reviewer's pointing out other instances of pattern mining being applied to genetic association studies. In the interest of citing the original literature where possible, we have referred to the suggested authors' 2021 paper and the Pounraja and Girirajan 2022 paper as examples of frequent-pattern mining in contrast to closed-pattern mining. The sentence now reads: "*Versions of closed-pattern mining (Pan et al, 2003; Terada et al, 2016; Relator et al, 2018; Yoshizoe et al, 2018) and a related approach called frequent-pattern mining (Pounraja and Girirajan 2022; Okazaki et al, 2021; Fang et al, 2012) have been applied in various genetic association studies to the learning of association rules between combinations of genes/genotypes and oligogenic disease; but to our knowledge none of these methods has been applied to haplotype analysis*" (p 5).

Page 4, line 11: The authors quote various papers as having applied closed pattern mining, but not all of these papers work with closed patterns. For example, the Pounraja and Girirajan 2022 paper works at the level of genes.

Thank you, we have modified and properly cited these papers. Please see our response to the previous comment.

Two lines down: The authors write "As it is haplotypes that are passed from parent to offspring" -- I don't understand the argument here. The only relevant fact seems whether alleles or genotypes are disease associated, no matter how they are passed from parent(s) to offspring.

Our intention here was to contrast the rare-variant hypothesis with the polygenic hypothesis and point out that under the former, there should be clustering of breast cancer risk in families due to rare variants or haplotypes of large effect. To clarify this point, we have referred to the genetic heterogeneity hypothesis mentioned earlier: "As it is haplotypes—not genotypes—that are passed from parent to offspring, it is pertinent *to look for the rare, possibly non-contiguous patterns of alleles that could underlie the genetic heterogeneity of inherited breast cancer risk.*" (p 5).

Page 7, line 2: "hit's" should be "hits", or "hit" if it was only a single hit.

We agree and have replaced "hit's" with "hit."

Page 7, line 5: "underlay" should be "underlie".

We agree and have adopted this change.

Page 10, line 2 up: "As at the" should probably just be "At the".

To clarify our use of “As,” we have modified this sentence to: “As at the chr11 locus, none of the original contiguous patterns *at these two loci* [i.e., the chr10 and chr16 loci] (minimum *P*-values 1.6×10^{-5} and 3.2×10^{-4}) met the inclusion threshold” (p 10).

Page 14, line 7: “linked to a GWAS hit” seems to refer to genetic linkage. The authors should not use “linkage” and “linked” (there is no genetic linkage here) but instead refer to (genetic) association.

We agree and have revised the sentences to: “*However, as we did not discover h1 in a region surrounding and in LD with a GWAS hit, we sought to explain the relationship between h1 and h38 instead. h1 and h38 were in moderate LD with each other ($r^2 = 0.59$, $D' = 1.0$) and in distinct LD with the GWAS hit, with the D' (r^2) of rs62237573 with h1 and h38 being 0.58 (0.068) and 0.90 (0.097), respectively*” (p 13).

Page 14, line 8: Should “explicate” be “explain”?

We agree and have adopted this change.

Page 18, “Methods” line 3: “consists in” should be “consists of”.

We would prefer to use “consists in” here, as we understand “consists of” to refer to component parts and “consists in” the essential characteristics of the object being discussed.

Page 19, last line: After all the high-flying theory, the authors mention here that they “only consider $\sigma = 2$ ”. They should describe in lay language what this means, and also make this clear early on in the manuscript.

We appreciate this suggestion and have modified the sentence to read: “*In this paper we only consider pairwise overlap, i.e., $\sigma = 2$. This restriction means that we potentially filter out risk-associated patterns which appear for the first time when three or more chromosomes are overlapped, but ensures that we miss no patterns shared by at least two*” (p 19).

July 24, 2023

RE: Life Science Alliance Manuscript #LSA-2023-02183-TR

Dr. William Letsou
New York Institute of Technology
Department of Chemical & Biological Sciences
Northern Blvd.
Theobald Hall, Rm. 425
Old Westbury, NY 11568

Dear Dr. Letsou,

Thank you for submitting your revised manuscript entitled "Refining the genetic risk of breast cancer with rare haplotypes and pattern-mining". We would be happy to publish your paper in Life Science Alliance pending final revisions necessary to meet our formatting guidelines.

- please add ORCID ID for the secondary corresponding author--they should have received instructions on how to do so
- please add the Twitter handle of your host institute/organization as well as your own or/and one of the authors in our system
- please incorporate the supplemental methods into the main manuscript Methods section, we do not have a size limit on this section

A. FINAL FILES:

B. MANUSCRIPT ORGANIZATION AND FORMATTING:

Sincerely,

Reviewer #1 (Comments to the Authors (Required)):

The authors have taken into account all my main comments and clearly improved the manuscript. I have no further comments.

July 26, 2023

RE: Life Science Alliance Manuscript #LSA-2023-02183-TRR

Dr. William Letsou
New York Institute of Technology
Department of Chemical & Biological Sciences
Northern Blvd.
Theobald Hall, Rm. 425
Old Westbury, NY 11568

Dear Dr. Letsou,

Thank you for submitting your Methods entitled "Refining the genetic risk of breast cancer with rare haplotypes and pattern-mining". It is a pleasure to let you know that your manuscript is now accepted for publication in Life Science Alliance. Congratulations on this interesting work.

DISTRIBUTION OF MATERIALS:

Again, congratulations on a very nice paper. I hope you found the review process to be constructive and are pleased with how the manuscript was handled editorially. We look forward to future exciting submissions from your lab.

Sincerely,
